# Incidence and Impacts of Inflammatory Bowel Diseases among Kidney Transplant Recipients: A Meta-Analysis

**DOI:** 10.3390/medsci8030039

**Published:** 2020-09-16

**Authors:** Panupong Hansrivijit, Max M. Puthenpura, Charat Thongprayoon, Himmat S. Brar, Tarun Bathini, Karthik Kovvuru, Swetha R. Kanduri, Karn Wijarnpreecha, Wisit Cheungpasitporn

**Affiliations:** 1Department of Internal Medicine, UPMC Pinnacle, Harrisburg, PA 17105, USA; hansrivijitp@upmc.edu; 2Department of Medicine, Drexel University College of Medicine, Philadelphia, PA 19129, USA; mmp356@drexel.edu; 3Division of Nephrology and Hypertension, Department of Medicine, Mayo Clinic, Rochester, MN 55905, USA; 4Department of Internal Medicine, University of Mississippi Medical Center, Jackson, MS 39216, USA; himmatbrar91@gmail.com; 5Department of Internal Medicine, University of Arizona, Tucson, AZ 85721, USA; tarunjacobb@gmail.com; 6Department of Medicine, Ochsner Medical Center, New Orleans, LA 70121, USA; karthik.kovvuru@ochsner.org (K.K.); svetarani@gmail.com (S.R.K.); 7Division of Gastroenterology and Hepatology, Department of Medicine, Mayo Clinic, Jacksonville, FL 32224, USA; wijarnpreecha.karn@mayo.edu

**Keywords:** inflammatory bowel diseases, kidney transplantation, meta-analysis, systematic reviews

## Abstract

Background: The incidence of inflammatory bowel diseases (IBD) and its significance in kidney transplant recipients is not well established. We conducted this systematic review and meta-analysis to assess the incidence of and complications from IBD in adult kidney transplant recipients. Methods: Eligible articles were searched through Ovid MEDLINE, EMBASE, and the Cochrane Library from inception through April 2020. The inclusion criteria were adult kidney transplant patients with reported IBD. Effect estimates from the individual studies were extracted and combined using the fixed-effects model when I^2^ ≤ 50% and random-effects model when I^2^ > 50%. Results: of 641 citations, a total of seven studies (*n* = 212) were included in the systematic review. The mean age was 46.2 +/− 6.9 years and up to 51.1% were male. The mean duration of follow-up was 57.8 +/− 16.8 months. The pooled incidence of recurrent IBD was 27.6% (95% CI, 17.7–40.5%; I^2^ 0%) while the pooled incidence of de novo IBD was 18.8% (95% CI, 10.7–31.0%; I^2^ 61.3%). The pooled incidence of post-transplant IBD was similar across subgroup analyses. Meta-regression analyses showed no association between the incidence of IBD and age, male sex, and follow-up duration. For post-transplant complications, the pooled incidence of post-transplant infection was 4.7% (95% CI, 0.5–33.3%; I^2^ 73.7%). The pooled incidence of graft rejection and re-transplantation in IBD patients was 31.4% (95% CI, 14.1–56.1%; I^2^ 76.9%) and 30.4% (95% CI, 22.6–39.5%; I^2^ 0%). Conclusion: Recurrent and de novo IBD is common among kidney transplant recipients and may result in adverse outcomes.

## 1. Introduction

Inflammatory bowel diseases (IBD), namely Crohn’s disease (CD) and ulcerative colitis (UC), are autoimmune gastrointestinal conditions with an estimated global prevalence of 6.8 million in 2017 [1]. Some immune-mediated conditions, such as autoimmune hepatitis and primary sclerosing cholangitis, have a strong association with IBD and are responsible for a large number of cases that require liver transplantations [2]. IBD prevalence and prognosis in liver transplant patients have been extensively studied over the years [3,4]. Verdonk et al. has shown that 44% of liver transplant patients had active IBD post-transplant. The cumulative risk for IBD after liver transplantation increased as these patients were followed for up to 10 years [2]. However, there are increasing concerns about whether a similar trend could be observed in other solid organ transplantations, especially in kidney transplant patients. To date, there is a scarcity of literature investigating the incidence of IBD post-kidney transplant and its impact on the kidney allograft.

In theory, IBD could be repressed in kidney transplant patients due to the immunosuppressive drugs such as corticosteroids and calcineurin inhibitors used for graft prophylaxis after transplantation [5]. These classes of medications are similar to those used in IBD treatment and thus could simultaneously dampen the autoimmune response responsible for clinical manifestations of IBD [6]. Thus, one could expect that the recurrence rate of IBD in kidney transplant patients is less than liver transplant patients. However, few studies describing the incidence of IBD or its clinical relevance in kidney transplant patients are present to date, leaving this theoretical assumption unsubstantiated. Thus, we aim to demonstrate the incidence of and complications from IBD in adult kidney transplant recipients through a systematic review and meta-analysis. The results of our current research would help provide directions for the planning of future clinical studies.

## 2. Materials and Methods 

### 2.1. Search Strategy

The manuscript follows the PRISMA (Preferred Reporting Items for Systematic Reviews and Meta-analysis) guidelines [7]. A systematic search was conducted through Ovid MEDLINE, EMBASE, and the Cochrane Library from database inception to April 2020 using the following search terms: (“inflammatory bowel diseases” OR “Crohn’s disease” OR “Ulcerative colitis”) AND (“kidney transplant” OR “renal transplant”) AND (“outcome” OR “graft function” OR “graft loss” OR “mortality” OR “death” OR “remission”). No language restrictions were applied. The search strategy for each database is summarized in Online Appendix A.

### 2.2. Inclusion Criteria

The eligibility of each study was determined by the following inclusion criteria: (1) the nature of the study is observational, a clinical trial, or conference abstract; (2) adult kidney transplant patients; (3) history of inflammatory bowel diseases in either pre-transplant or post-transplant was described. Case reports, case series, review articles, or articles concerning pediatric patients were excluded. Study eligibility was independently evaluated by two investigators (P.H. and M.P.). Any disagreements were resolved by mutual consensus among all authors. The quality of each study was appraised using the Newcastle–Ottawa quality scale [8], which assesses six components, including (1) representativeness of the subjects; (2) ascertainment of the exposure; (3) demonstration of the outcome of interest was not present at the start of the study; (4) assessment of outcome; (5) follow-up duration period was long enough for the outcome to occur; and (6) adequate follow-up duration.

### 2.3. Review Process and Data Extraction

The abstracts and titles of all citations were first screened by two investigators (P.H. and M.P.) before going through a full-text review. The full-text of each remaining article was reviewed to determine the eligibility to be included in the systematic review and meta-analysis. A standardized data collection form was used to extract the following information from the included studies: first author’s name, year of publication, year of study, the country where the study was conducted, study design, subject(s), sample size, age, male sex, ethnicity, identifying symptom(s), such as diarrhea, abdominal pain or bleeding/anemia, immunosuppressive regimen, use of anti-tumor necrosis factor-α (TNF-α) therapy, reported outcome(s), and follow-up duration.

### 2.4. Measurements

The incidence of IBD, the rejection rate, and the infection rate that resulted from meta-analyses were reported as a percentage along with a 95% confidence interval (CI) and a forest plot. Rejections were diagnosed by kidney biopsy histological criteria compatible with either antibody-mediated rejection or cellular rejection. Infection rate accounts for any significant viral, bacterial, or fungal infections requiring hospitalization. Descriptive statistics were presented as a percentage for categorical data and as a mean ± standard deviation (S.D.) for continuous data.

### 2.5. Subgroup Analysis, Meta-Regression Analysis, and Publication Bias

Subgroup analyses were performed by classifying the included studies based on disease nature (recurrent IBD vs. de novo IBD), study year (≤2015 vs. >2015), country of origin (USA vs. others), study design (prospective vs. retrospective), and ethnicity (white vs. others). A Mixed-effect model of analysis was used. Meta-regression analyses were performed to establish an association between study characteristics, such as age, male sex, and duration of follow-up, and the incidence of IBD. The results of meta-regression analyses were presented in scatterplots and raw spreadsheets. Publication bias was evaluated using Egger’s regression intercept. A value of less than 0.05 was considered significant for potential publication bias.

### 2.6. Statistical Analysis

All statistical analyses were performed by the Comprehensive Meta-analysis version 3 software (Eaglewood, NJ, USA) and SPSS version 23.0 (IBM Corp., Armonk, NY, USA). Statistical heterogeneity of studies was assessed using Cochran’s Q-test, which was supplemented by I^2^ statistics. I^2^ statistics quantify the proportion of the total variation across studies. A value of ≤25% represents insignificant heterogeneity, 25–50% represents low heterogeneity, 50–75% represents moderate heterogeneity, and >75% represents high heterogeneity [9]. For analyses with I^2^ ≤ 50%, the results were derived from the fixed-effects model. The random-effects model of analysis was used to minimize the heterogeneity and external variance. Thus, the random-effects model was used if the I^2^ index generated by the fixed-effects model was greater than 50% [10]. A p-value of less than 0.05 represents statistical significance.

## 3. Results

### 3.1. Study Characteristics

Of 641 citations, a total of seven studies consisting of 212 subjects were included in the systematic review and meta-analysis. Figure 1 illustrates the flowchart of the literature search and study selection. The risk of bias summary is presented in Appendix A. Studies dated from 2013 to 2019. The origin of studies was France (*n* = 2), the United States (*n* = 2), Germany (*n* = 1), Poland (*n* = 1), and Israel (*n* = 1). The study design was retrospective (57.1%) and prospective (42.9%). All studies included subjects undergoing kidney transplantation regardless of previous history of IBD. The mean age was 46.2 ± 6.9 years and up to 51.1% were male. The identifying symptoms were reported in only two studies [11,12], including diarrhea, abdominal pain, and anemia/bleeding. The mean duration of follow-up was 57.8 ± 16.8 months. Table 1 illustrates the study characteristics and outcomes.

### 3.2. Recurrent IBD

Of three cohorts involving 58 subjects, the pooled incidence of recurrent IBD was 27.6% (95% CI, 17.7–40.5; I^2^ = 0%). This result was analyzed using the fixed-effects model. The forest plot from the meta-analysis of the pooled incidence of recurrent IBD is illustrated in Figure 2A.

### 3.3. De Novo IBD

Of four cohorts including 186 subjects, the pooled incidence of de novo IBD was 18.8% (95% CI, 10.7–31.0; I^2^ = 61.3%). This analysis was done using the random-effects model. The forest plot from the meta-analysis of the pooled incidence of de novo IBD is depicted in Figure 2B.

### 3.4. Post-Transplant Infection

Of four cohorts consisting of 116 subjects, the pooled incidence of post-transplant infection was 4.7% (95% CI, 0.5–33.3; I^2^ = 73.7%). The Random-effects model was used to conduct the analysis. The forest plot from the meta-analysis of the pooled incidence of post-transplant infection is illustrated in Figure 2C.

### 3.5. Graft Rejection

From three cohorts including 62 patients, the pooled incidence of graft rejection in IBD patients was 31.4% (95% CI, 14.1–56.1; I^2^ = 76.9%). The analysis was performed using a random-effects model. The forest plot from the meta-analysis of the pooled incidence of graft rejection in IBD patients is illustrated in Figure 3A.

### 3.6. Re-Transplantation

From three cohorts with a total of 58 patients, the pooled incidence of re-transplantation in IBD patients was 30.4% (95% CI, 22.6–39.5; I^2^ = 0%). This result was analyzed using the fixed-effects model. The forest plot from the meta-analysis of the pooled incidence of re-transplantation in IBD patients is illustrated in Figure 3B.

### 3.7. Subgroup Analyses

The results of the subgroup analyses are depicted in Table 2. Here, we analyzed the pooled incidence of post-transplant IBD based on the study characteristics. In brief, we found that the pooled incidence of post-transplant IBD was similar after adjusting for disease nature (recurrent IBD vs. de novo IBD), study year (≤2015 vs. >2015), country of origin (USA vs. others), study design (prospective vs. retrospective), and ethnicity (white vs. others). We applied the mixed-effects model to conduct the subgroup analyses to minimize inter-study variance.

### 3.8. Meta-Regression Analyses

Meta-regression analyses were used to delineate the association between dependent variables and the outcome of interest—in this case, the incidence of post-transplant IBD. Here, we found that age, male sex, or the duration of follow-up were not associated with the incidence of post-transplant IBD. The results of meta-regression analyses are available in Online Appendix A. The scatterplots between log event rate (IBD incidence) and dependent variables (age, male sex, and follow-up duration) are illustrated in Figure 4.

#### Evaluation for Publication Bias

The applicability of the funnel plot was limited due to an insufficient number of total included studies [18]. The Egger’s regression intercept for the analysis of the pooled incidence of recurrent IBD, de novo IBD, and post-transplant infection was 0.757, 0.096, and 0.309, respectively. Similarly, the Egger’s regression intercept for the analysis of the pooled incidence of graft rejection and re-transplantation in IBD patients was 0.601 and 0.481, respectively. These values indicated no potential publication bias.

## 4. Discussion

With a mean follow-up close to 5 years, we demonstrated that the incidence of recurrent IBD and de novo IBD was 27.6% and 18.8%, respectively. Although it is promising that the incidence of recurrent IBD is higher than de novo IBD, this difference did not reach statistical significance, possibly due to a lack of statistical power. In 2006, a retrospective study of 91 liver transplant patients from the University of Nebraska Medical Center showed that up to 65% of IBD recurred post-transplant and 19% had de novo IBD [2]. It is expected that recurrent IBD is less common in kidney transplant recipients because IBD is more associated with hepatic manifestations. Although IBD can result in renal amyloidosis, the incidence is rare—at least 1% of IBD patients will develop systemic AA amyloidosis [19]. The reported incidence of recurrent IBD was seemingly lower than non-transplant patients. One study showed that 55% of IBD patients experienced a relapse with a median time to relapse of 32 and 18 months for CD and UC, respectively [20]. Moreover, whether the behavior of recurrent IBD resembles pre-transplant IBD is undetermined.

Continuous immunosuppression in kidney transplant patients is believed to play a major role in preventing IBD relapses. The results from a large Scandinavian meta-analysis suggested that switching from tacrolimus-based to cyclosporine-based anti-rejection regimen following liver transplantation was associated with a more favorable outcome toward reducing IBD activity [21]. In this study, a dual treatment of tacrolimus and mycophenolate mofetil was a significant risk factor for increased IBD activity after liver transplantation. Similar findings were also demonstrated by other groups of researchers [2,22]. It is worth noting that tacrolimus is superior to cyclosporine in improving graft survival and preventing graft rejection after kidney transplantation [23]. Switching from tacrolimus to cyclosporine, aiming to prevent IBD relapses, cannot be recommended due to a lack of substantiated evidence. Anti-TNF-α therapy was an effective therapeutic option in patients with refractory IBD in several clinical trials and the use of anti-TNF-α has been extended to solid organ transplant patients [24,25,26]. Moreover, a previous in vivo study pinpointed the essential role of TNF-α in allograft rejection [27]. Hence, anti-TNF-α therapy is a promising alternative treatment for patients with acute rejection and active IBD. The benefits of anti-TNF-α therapy in solid organ transplant patients should be demonstrated in future randomized clinical trials.

As suggested by Verdonk et al., long term use of calcineurin-inhibitors (CNI), such as tacrolimus or cyclosporin A, may also contribute to increased IBD activity and account for both recurrent or de novo IBD [2]. The proposed mechanisms of CNI-induced IBD seem to involve regulatory T cells and directly influence intestinal permeability. Tacrolimus was shown to increase the intestinal permeability in liver transplant recipients, leading to increased exposure of luminal antigens to stimulate the intestinal immune system [28]. Moreover, CNIs reduce inflammation by suppressing IL-2 dependent proliferation of regulatory T cells [29,30,31]. Although this process seems therapeutic, prolonged suppression of regulatory T cells could lead to less activation-induced cell death (AICD) of T cells. This mechanism makes T cells more resistant to apoptosis [32,33,34]. More importantly, these mechanisms may explain the observation that patients with rejection tend to develop IBD as these patients are aggressively immunosuppressed.

Here, we showed that age and sex were not associated with post-transplant IBD. This finding is different from what has previously been described in non-transplant patients. Young age at diagnosis and female sex are clinical predictors for relapses in IBD patients [35]. Interestingly, our findings also suggested that the duration of follow-up was not associated with post-transplant IBD. This could reflect the fact that post-transplant IBD usually occurred early, at least within 5 years. This hypothesis corresponds with the pre-transplant data in which most relapsed cases of IBD occurred within the first 3 years after remission [20]. It is also possible that the practicality of our results is limited due to the small pooled sample size. Additional studies are required to demonstrate the risk factors of IBD relapses following kidney transplantation.

The pooled incidence of post-kidney transplant infections from our analysis was 4.7%, which was lower than other cohorts. One retrospective study reported that up to 25% of patients had at least one episode of nosocomial infection. Patients with mammalian target of rapamycin (mTOR) inhibitors showed a significantly higher infection rate than those receiving CNIs [36]. However, potential reporting bias should be taken into consideration when interpreting our findings as the infection rate was not the primary outcome in most included studies. Additionally, it is possible that included studies tended to report serious infection events but rather ignored mild infectious episodes—making the overall infection rate lower than expected.

The incidence of rejection and re-transplantation in IBD patients was 31.4% and 30.4%, respectively. Based on the United States Renal Data System (USRDS), 14.3% of patients on the kidney transplant waitlist were re-transplant candidates [37]. However, due to insufficient data, it is unclear what the causes of graft loss in our patients were. Additional studies evaluating patients with IBD who required second kidney transplantation are encouraged. Rejection episodes may result from under-suppression of the immune system or associated infections. A newer study has shown that the evolution of immunosuppressants has decreased the incidence of acute rejection and improved graft survival after kidney transplantation [38]. In this cohort, the incidence of acute rejection episodes was 68.7% on azathioprine era and reduced to 38.2% on cyclosporine A and 11.4% on tacrolimus. However, although the immunosuppressive regimens were not reported in all included studies, it is assumed that most patients received cyclosporine A or tacrolimus given the study year and previous rejection profile from Marcen et al. [38]. Again, physicians are encouraged to weigh the risk of rejection vs. risk of IBD relapse when considering one CNI over the other for kidney transplant patients with a history of IBD. Additional large-scale observational studies are helpful.

To date, this is the first meta-analysis to describe the incidence of IBD and its complications in kidney transplant recipients. However, some limitations are to be applied. A small sample size and unavailability of the full data from some included articles may attenuate the power of the study. With this limitation, we were unable to differentiate patients with Crohn’s disease and ulcerative colitis to be analyzed independently. Similarly, the odds ratio cannot be calculated due to insufficient data. Moreover, given the observational nature, the included studies may be subjected to selection bias. More importantly, kidney transplant patients are complex, thus they are likely susceptible to potential confounding factors, such as maintenance immunosuppressive regimens, and induction regimens. Hence, the interpretation and applicability of our findings should be performed with caution.

## 5. Conclusions

In conclusion, IBD in kidney transplant recipients is common with an incidence of 27.6% for recurrent IBD and 18.8% for de novo IBD. The rejection rate among kidney transplant patients with IBD was 31.4%. Large-scale observational studies describing the clinical impacts of IBD in kidney transplant patients are needed. Moreover, our study encouraged future clinical trials of anti-TNF-α therapy in kidney transplant patients with IBD as this agent might reduce IBD relapse as well as prevent graft rejection.

## Figures and Tables

**Figure 1 medsci-08-00039-f001:**
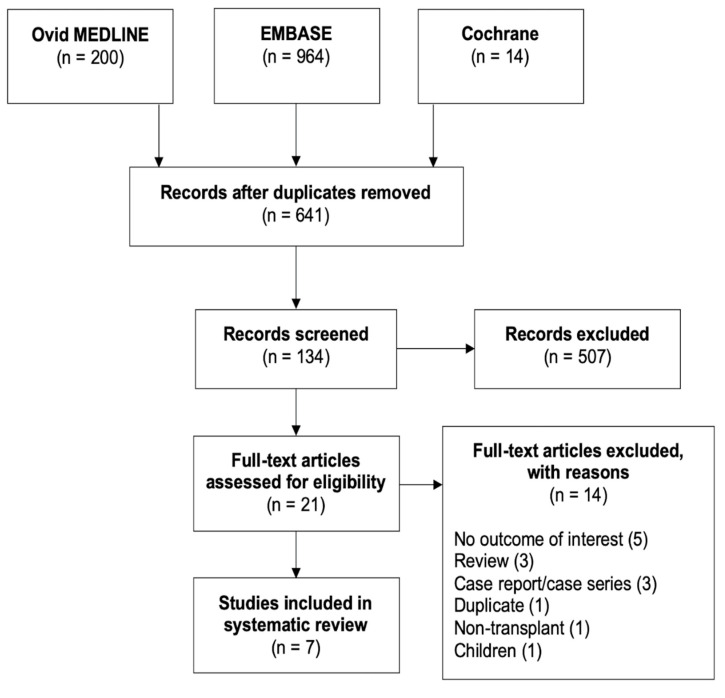
Preferred Reporting Items for Systematic Reviews and Meta-analysis (PRISMA) flowchart of article search and selection.

**Figure 2 medsci-08-00039-f002:**
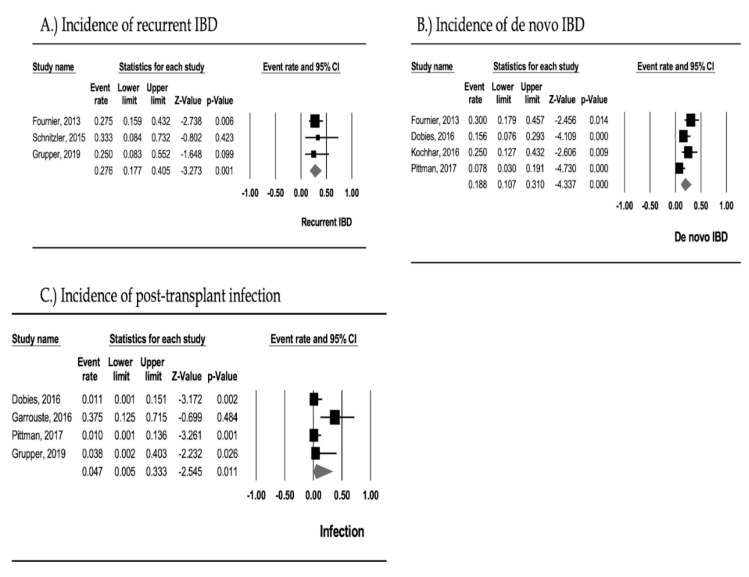
Forest plots of meta-analysis on (**A**) the pooled incidence of recurrent inflammatory bowel diseases (I^2^ 0%; Egger’s intercept 0.757), (**B**) the pooled incidence of de novo inflammatory bowel diseases (I^2^ 18.8%; Egger’s intercept 0.096), (**C**) the pooled incidence of post-transplant infection (I^2^ 73.3%; Egger’s intercept 0.309).

**Figure 3 medsci-08-00039-f003:**
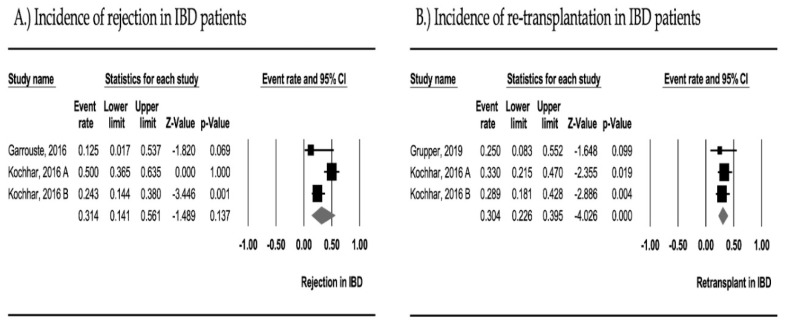
Forest plot of meta-analysis on (**A**) the pooled incidence of graft rejection in inflammatory bowel disease patients (I^2^ 76.9%; Egger’s intercept 0.601) and (**B**) the pooled incidence of re-transplantation in inflammatory bowel disease patients (I^2^ 0%; Egger’s intercept 0.481).

**Figure 4 medsci-08-00039-f004:**
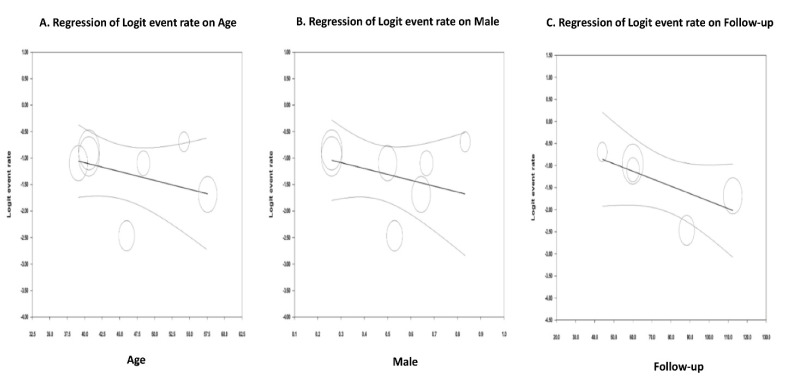
Scatter plot from meta-regression analysis on (**A**) age, (**B**) male sex, and (**C**) follow-up.

**Table 1 medsci-08-00039-t001:** Included studies.

Study	Country	Design	N	Subjects	Age, yr	Male, %	Regimen	Anti-TNF Pre-KTx	Diarrhea, %	Abdominal Pain, %	Anemia or Bleeding, %	Outcomes	Follow-Up
Fournier, 2013 [13]	France	R	40	IBD + KTx	40.6	26	-	-	-	-	-	Recurrent IBD: 27.5% De novo IBD: 30.0% Median delay of flare-up: 17 mo in recurrent IBD, 91 mo in de novo IBD CMV: 36.0% in recurrent IBD, 5.9% in no recurrent IBD	-
Schnitzler, 2015 [14]	Germany	P	6	IBD + KTx	54.2	83.3	66.7% CsA, 33.3% Tac	0%	-	-	-	Recurrent IBD: 33.3%	112.5 mo
Dobies, 2016 [11]	Poland	P	45	KTx	57.6	64.4	-	-	40	37.8	15.6	De novo IBD: 15.6% Infection: 0%	-
Garrouste, 2016 [15]	France	P	8	IBD + KTx + anti-TNF	46.5	50	-	100%	-	-	-	Infection: 37.5% Cancer: 37.5% Acute rejection: 12.5%	88.3 mo
Kochhar, 2016 [16]	USA	R	50	KTx	39.1	50	-	-	-	-	-	De novo IBD: 25% Graft rejection in de novo IBD (50.0%) vs. non-de novo IBD (24.3%) Re-transplantation in de novo IBD (33%) vs. non-de novo IBD groups (28.9%), *p* = 0.32	60 mo
Pittman, 2017 [12]	USA	R	51	KTx	46	53	-	-	57	12	27	De novo IBD: 7.8% (25% CD, 75% UC) Infection: 0%	44 mo
Grupper, 2019 [17]	Israel	R	12	IBD + KTx (41.7% UC, 58.3% CD)	48.4	66.7	91.7% CNI, MMF, Pred	50%	-	-	-	Recurrent IBD: 25% (33.3% UC, 66.7% CD) Re-transplantation: 16.7% Infection: 0%	60.1 mo

Abbreviations: CD, Crohn’s disease; CMV, cytomegalovirus; CNI, calcineurin inhibitor; CsA, cyclosporine A; IBD, inflammatory bowel disease; KTx, kidney transplant; MMF, mycophenolate mofetil; P, prospective; Pred, prednisone/prednisolone; R, retrospective; Tac, tacrolimus; TNF, tumor necrosis factor; UC, ulcerative colitis.

**Table 2 medsci-08-00039-t002:** Subgroup analyses.

Subgroup Analyses				
Subgroup	N	Incidence	95% CI	
Year				
≤2015	3	29.1%	20.5–39.5	
>2015	4	16.9%	10.0–27.1	Q = 3.267, *p* = 0.071
Country				
USA	2	14.8%	4.3–39.9	
Others	5	25.1%	18.6–33.0	Q = 0.854, *p* = 0.356
Study design				
Prospective	2	18.5%	9.4–33.2	
Retrospective	5	22.6%	14.8–33.1	Q = 0.281, *p* = 0.596
Disease				
Recurrent IBD	3	27.6%	17.7–40.5	
De novo IBD	4	18.8%	10.7–31.0	Q = 1.239, *p* = 0.096
Ethnicity				
White	4	25.1%	18.3–33.5	
Others	3	17.3%	7.7–34.4	Q = 0.849, *p* = 0.357

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
