# Peer review of "Incidence and Impacts of Inflammatory Bowel Diseases among Kidney Transplant Recipients: A Meta-Analysis"

_medsci, 2020, doi:10.3390/medsci8030039_

Round 1

Reviewer 1 Report

In contrast to liver transplantation, renal transplantation (KTx) is rarely considered a condition that may be complicated with inflammatory bowel disease (IBD). Surprisingly, the Authors in their meta-analysis show that IBD is an underappreciated problem in kidney transplant recipients; the frequency of reccurent IBD is 27.6%, and de novo IBD is 18.8% in this group. The article is, therefore, an important indication that IBD after KTx shoud become a condition that is more often considered in clinical settings as well as in research on clinical outcome after KTx. The paper is well designed, and written. I suggest to accept it in a present form.

Author Response

Response to Reviewer #1

Reviewer 1:

In contrast to liver transplantation, renal transplantation (KTx) is rarely considered a condition that may be complicated with inflammatory bowel disease (IBD). Surprisingly, the authors in their meta-analysis show that IBD is an underappreciated problem in kidney transplant recipients; the frequency of recurrent IBD is 27.6%, and de novo IBD is 18.8% in this group. The article is, therefore, an important indication that IBD after KTx should become a condition that is more often considered in clinical settings as well as in research on clinical outcome after KTx. The paper is well designed, and written. I suggest to accept it in a present form.

Response: We thank you for reviewing our manuscript and for your critical evaluation. Thank you so much for your kind comments. We were surprised that the incidence of recurrent IBD was close to 30% in KTx patients. More importantly, the incidence of de novo IBD in KTx patients was also about 20%. This has emphasized our suspicion that aggressive immunosuppression in KTx patients might increase the likelihood of developing IBD. Thank you.

Reviewer 2 Report

Interesting analysis of the role of IBD on clinical outcomes in kidney recipients.

Please provide additional information about definition of analysed complications including rejection and infection.

Author Response

Response to Reviewer #2

Interesting analysis of the role of IBD on clinical outcomes in kidney recipients.

Please provide additional information about definition of analysed complications including rejection and infection.

Response: We thank you for reviewing our manuscript and for your critical evaluation. Thank you for your comments. We agree with the reviewer. We have added the definitions of ‘infection’ and ‘rejection’ under Methods section (Line 97-101), “The incidence of IBD, the rejection rate, and the infection rate resulted from meta-analyses were reported in percentage along with a 95% confidence interval (CI) and the forest plot. Rejections were diagnosed by kidney biopsy histological criteria compatible with either antibody-mediated rejection or cellular rejection. Infection rate accounts for any significant viral, bacterial or fungal infections requiring hospitalization.”

All authors thank the Editors and reviewers for their valuable suggestions. The manuscript has been improved considerably by the suggested revisions! 

Reviewer 3 Report

The manuscript entitled " Incidence and Impacts of Inflammatory Bowel Diseases among Kidney Transplant Recipients: A Meta-analysis" describes the common presence of IBD in kidney transplant patients.
My major criticism towards the manuscript is that the use of drugs that affect the cell division in the gut may adequately explain the great incidence of gastrointestinal symptoms in transplanted patients. The authors focus on the role of the immune system, whereas a direct effect of immunosuppressive drugs on the bowel seems more appropriate. It is even possible that the IBD post-transplant is unrelated from the IBD in the general population, and it has an iatrogenic origin.
This consideration would also adequately explain the reported association between IBD and rejection: patients with rejection usually are treated with greater doses of immunosuppressors, which might cause IBD.

Author Response

Response to Reviewer #3

The manuscript entitled "Incidence and Impacts of Inflammatory Bowel Diseases among Kidney Transplant Recipients: A Meta-analysis" describes the common presence of IBD in kidney transplant patients. My major criticism towards the manuscript is that the use of drugs that affect the cell division in the gut may adequately explain the great incidence of gastrointestinal symptoms in transplanted patients. The authors focus on the role of the immune system, whereas a direct effect of immunosuppressive drugs on the bowel seems more appropriate. It is even possible that the IBD post-transplant is unrelated from the IBD in the general population, and it has an iatrogenic origin. This consideration would also adequately explain the reported association between IBD and rejection: patients with rejection usually are treated with greater doses of immunosuppressors, which might cause IBD.

Response: We thank you for reviewing our manuscript and for your critical evaluation. Thank you for helpful suggestions. We agreed with the reviewer. We have additionally reviewed the literature, and agreed with this important suggestion. Thus, we have additionally added discussion as reviewer’s sugegestion.

 “As suggested by Verdonk et al., long term use of calcineurin-inhibitors (CNI), such as tacrolimus or cyclosporin A may also contribute to increased IBD activity and account for both recurrent or de novo IBD (2). The proposed mechanisms of CNI-induced IBD seem to involve regulatory T cells and direct influence to the intestinal permeability. Tacrolimus was shown to increase the intestinal permeability in liver transplant recipients leading to increased exposure of luminal antigens to stimulate intestinal immune system (27). Moreover, CNIs reduce the inflammation by suppressing IL-2 dependent proliferation of regulatory T cells (28-30). Although, this process seems therapeutic, prolonged suppression of regulatory T cells could lead to less activation-induced cell death (AICD) of T cells. This mechanism makes T cells more resistant to apoptosis (31-33). More importantly, these mechanisms may explain the observation that patients with rejection tend to develop IBD as these patients are aggressively immunosuppressed.”

All authors thank the Editors and reviewers for their valuable suggestions. The manuscript has been improved considerably by the suggested revisions! 
